

# Fermentation characteristics of bedded pack barn dairy cattle manure on methane yield, carbon, and nitrogen content in solid-state anaerobic digestion

Yongjun Choi[1], Sangrak Lee[1], Duck-Min Ha[2], Youngjun Na[3] and Doo-Hwan Kim[2]

[1] Department of Animal Science and Technology, Konkuk University, Seoul, Seoul, South Korea
[2] Division of Animal Science, Agri-Food Bio Convergence Institute, Gyeongsang National University, Jinju, Gyeongsangnam-do, South Korea
[3] Animal Data Lab, Antller Inc., Seoul, Seoul, South Korea

## ABSTRACT

This study aimed to estimate the fermentation characteristics of bedded pack barn dairy cattle manure (BDCM) in terms of methane yield, fibrous material, and nitrogen content in batch solid-state anaerobic digestion (SSAD). SSAD was performed in triplicate using a 1,400 ml polypropylene bottle at a constant temperature of 39 °C until less than 1% methane was produced. The cumulative methane content of BDCM was 142.5 N mL/g volatile solids (VSs). The methane content rapidly increased for 18 days, reaching $63.4 \pm 4.6\%$ until the end of the experiment. The ultimate biodegradability and total VS removal of BDCM were 23.1 and 19.0%, respectively. The slopes of the non-fibrous and hemicellulose carbon fractions, and acid detergent insoluble carbon by digestion time were $-0.174$ ($p < 0.001$), $-0.141$ ($p = 0.003$), and $-0.051$ ($p < 0.001$), respectively. The non-fibrous and hemicellulose nitrogen fraction contents quadratically decreased during SSAD ($p = 0.001$ and $p = 0.008$). No significant decrease was observed in the acid detergent insoluble nitrogen content ($p = 0.840$). The results of the present study provide basic data on the digestion characteristics of BDCM and could help determine fermentation conditions in the anaerobic digestion of BDCM.

## INTRODUCTION

Anaerobic digestion is more beneficial than composting (AD, aerobic digestion) due to the production of methane as fuel and low energy consumption (*Fricke, Santen & Wallmann, 2005*). In South Korea, the anaerobic digestion of livestock manure is a rare practice, and is only carried out in wet states using mainly swine manure. According to the data provided by the *South Korean Ministry of Environment (2020)*, more than 250 units of 100 ton/day scale biogas plants are theoretically required to recycle all livestock manure in South Korea. Recently, climate change has elicited a growing interest in renewable energy; however, the study of basic technology is necessary to promote anaerobic digestion technology.

Corresponding author
Doo-Hwan Kim, dhkim61@gnu.ac.kr

Anaerobic digestion is classified on the basis of moisture content of the substrates. The treatment of substrates with 80–85% moisture is generally referred to as solid-state anaerobic digestion (SSAD), whereas wet-state anaerobic digestion (WSAD) is performed at a moisture content of over 95%. Therefore, SSAD has an advantage in terms of digesting more substrates in the same volume than WSAD. However, it has several disadvantages including high mechanical overload for stirring, slow startup phase, and long hydraulic retention time compared with WSAD. Many studies have focused on enhancing the anaerobic digestion process in various ways (*Yenigün & Demirel, 2013*; *Mao et al., 2015*). In South Korea, most of the dairy cattle are bred in bedded pack barns using sawdust as the most common bedding material. The mixing of sawdust and cattle manure may affect the characteristics of SSAD. Previous studies have assessed the effect of moisture content and substrate-to-inoculum ratio on methane production from a mixture of cattle manure and sawdust using SSAD (*SeungHun et al., 2017*; *Kim et al., 2018*). However, studies on carbon and nitrogen of cattle manure mixed with sawdust are rare during SSAD.

Anaerobic digesters are broadly divided into batch and continuous types. Batch AD is a process in which all the substrates are supplied at the start, followed by product recovery at the end of the process. In contrast, feeding substrates are supplied and excreted daily in the same amount during the continuous AD process, resulting in a constant amount of digestate in the digester unlike that in batch AD. Generally, batch AD is performed to determine the maximum methane yield, calculate the hydraulic retention time, and determine various digestion characteristics for the setup of the control point before continuous AD operation.

Therefore, the present study aimed to evaluate the fermentation characteristics of bedded pack barn dairy cattle manure (BDCM), and to estimate the change in carbon and nitrogen content of its carbon fraction during batch SSAD.

## MATERIALS & METHODS

### Dairy cattle manure

The BDCM was collected at a facility with 0.02 head/m$^2$ density of dairy cattle and 3.03 kg/m$^2$ bedded sawdust in a cow farm in *Yangpyung* province (Latitude, 37.4614204; longitude, 127.5285745). The moisture content and volatile solids in sawdust were recorded as 36.1 ± 0.2% and 98.6 ± 0.1%, respectively. The BDCM characteristics considered in this study are listed in Table 1. Manure samples were randomly collected from at least five or more points and cow barns. Manure was collected eight times or more from each sampling point (*Dou et al., 2001*). The collected samples were classified using the VDI4630 method (*Standard, 2006*) and stored at −20 °C until further analysis.

### Batch SSAD

The SSAD tests were carried out in triplicate using an approximately 1,400 mL polypropylene batch digester equipped with a 2 L gas bag at a constant temperature of 39 °C. The BDCM stored at 4 °C was pre-incubated at a temperature up to 39 °C before AD. The anaerobic state was established using N$_2$ gas inside the bottles which were kept in a shaking incubator (IS-971R; Jeiotech Co., Korea) until less than 1% CH$_4$ was produced. The substrate was supplied in the digester at 462.3 ± 4.9 g (wet basis)

**Table 1** Chemical characteristics of the bedded pack barn dairy cattle manure used in the solid-state anaerobic digestion ($n = 3$).

| Item | BDCM (Mean ± SD) |
| --- | --- |
| TS (%, w.b.) | 20.3 ± 0.0 |
| VS (%, d.b.) | 83.3 ± 0.4 |
| Cellulose (%, d.b.) | 18.2 ± 0.2 |
| Hemicellulose (%, d.b.) | 19.8 ± 0.2 |
| Lignin (%, d.b.) | 17.5 ± 0.3 |
| Carbon (%, d.b.) | 43.3 ± 0.2 |
| Nitrogen (%, d.b.) | 2.8 ± 0.0 |

**Notes.**
TS, total solids; VS, volatile solids; w.b., Wet basis; d.b., Dry basis; BDCM, bedded pack barn dairy cattle manure.

weight and moisture content adjusted to 82% using a mixture of distilled water, inoculum, and medium. Additionally, inoculum and medium were added at 0.88 and 0.44 g/g of total solids, respectively. The inoculum used in the study could digest cattle manure fed various substrates, and was collected from the batch type mesophilic anaerobic digester at 39 °C, which produced about 500 N mL $CH_4$/g volatile solids (VSs)/d of biogas with 60% methane concentration The medium used for AD was prepared using the method described by *Angelidaki & Sanders (2004)* (Table 2). The final pH of the medium was adjusted to 7.1 using $CO_2$ gas, and stored at 39 °C before AD. All assays were performed once every three days to measure gas production during the experimental period. Sampling was performed using a 50 mL gas-tight glass syringe (1001 SL SYR; Hamilton Co., Reno, USA) to measure gas production and composition every three days for 72 days. The sampling for recording pH, ammonia nitrogen, volatile fatty acids (VFA), and chemical composition was performed on the 0, 3, 6, 12, 18, 24, 30, 36, 48, 60, and 72nd days.

## Analytical method

Total and volatile solids were determined using the method described by *APHA APH (2018)*. Elemental analyses of C, H, N, S, and O were performed using an elemental analyzer (EA 1110, CE Instruments, Italy), and pH was determined using a pH meter (Orion 420A+; Thermo Electron Co., Waltham, MA, USA). Ammonia nitrogen content was determined using the method described by (*Rice, Baird & Eaton, 2017*) and an ELISA reader (Synergy 2, Biotek Co., USA) according to Eq. (1) (*Drosg, 2013*).

$$NH_3 \, (\text{mg/L}) = \frac{NH_4 - N}{1 + 10^{(0.0925 + 2728.795/T - pH)}} \tag{1}$$

T: 273.15 K + gas temperature (° C)

Gas production was determined daily using a 50 ml glass syringe. The amount of gas produced was calibrated to standard temperature and pressure (STP 0 °C, 1 atm) considering the temperature-dependent volume using the following calibration (Eq. (2)):

$$V = V_{\text{at } T°C} \times \frac{273}{(273 + T)} \times \frac{(P - P_W)}{760} \tag{2}$$

where, V is the gas production at 0 °C and 1 atm, $V_{\text{at } T°C}$ is the gas production at T°C, T is the temperature at the time of the volume measurement, P is the pressure at the time

**Table 2 Compounds of anaerobic media for solid-state anaerobic digestion.**

| Compound | Chemical formula | Concentration (mg/L) |
| --- | --- | --- |
| Sodium chloride | $NaCl$ | 100 |
| Magnesium chloride hexahydrate | $MgCl_2 \cdot 6H_2O$ | 100 |
| Calcium chloride dihydrate | $CaCl_2 \cdot 2H_2O$ | 50 |
| Dipotassium phosphate trihydrate | $K_2PO_4 \cdot 3H_2O$ | 400 |
| Cysteine hydrochloride | $C_3H_7NO_2S \cdot HCl$ | 500 |
| Sodium bicarbonate | $NaHCO_3$ | 2600 |
| Resazurin | $C_{12}H_6NNaO_4$ | 0.5 |
| Iron (II) chloride | $FeCl_2 \cdot 4H_2O$ | 2.0 |
| Boric acid | $H_3BO_3$ | 0.05 |
| Zinc chloride | $ZnCl_2$ | 0.05 |
| Copper (II) chloride | $CuCl_2 \cdot 2H_2O$ | 0.038 |
| Manganese (II) chloride | $MnCl_2 \cdot 4H_2O$ | 0.05 |
| Ammonium molybdate tetrahydrate | $(NH_4)6Mo7O_{24} \cdot 4H_2O$ | 0.05 |
| Aluminum chloride | $AlCl_3$ | 0.05 |
| Cobalt (II) chloride hexahydrate | $CoCl_2 \cdot 6H_2O$ | 0.05 |
| Nickel (II) chloride | $NiCl_2 \cdot 6H_2O$ | 0.092 |
| Ethylenediamine tetra-acetic acid | $C_{10}H_{16}N_2O_8$ | 0.5 |
| Concentrated HCl | $HCl$ | 1.0 |
| Sodium selenite pentahydrate | $Na_2SeO_3 \cdot 5H_2O$ | 0.1 |

of the volume measurement, and $P_w$ is the saturated water vapor pressure at T°C. The gas composition was determined by using gas chromatography (HP 6890; Hewlett-Packard Co., Palo Alto, CA, USA) with a thermal conductivity detector. The sample (0.2 ml) containing helium as the carrier gas was injected into the column at a temperature of 60 °C and flow rate of 1.5 ml/min. The sample gas concentration was calibrated using a standard gas mixture consisting of 40% $CH_4$–60% $CO_2$ and 60% $CH_4$–40% $CO_2$.

The amount of methane produced from the serum bottle was calibrated using the calibration equation (Eq. (3)) presented by *Shin (2002)*:

$$C_{CH_4(R)} = C_{CH_4} \times \frac{100}{(C_{CH_4} + C_{CO_2})} \tag{3}$$

where, $C_{CH_4(R)}$ is the calibrated methane concentration (%), $C_{CH_4}$ is the measured methane concentration (%), and $C_{CO_2}$ is the measured carbon dioxide concentration (%). The volatile solid reduction rate was determined by the equation described by *Kim et al. (2018)*.

## Ultimate biodegradability and biomass removal

Total volatile solids (TVSs) consisted of biodegradable volatile solids (BVSs) and non-BVSs, and the TVS residue ($TVS_e$) after degradation was calculated using Eqs. (4)–(6). The ultimate biodegradability was calculated by plotting the ratio of $TVS_e$ and initial TVS before degradation ($TVS_0$) on the $Y$-axis and reciprocal of the operating time (1/time) on the $X$-axis according to the method described by *Kim et al. (2018)*.

Biomass removal (BMR) = $CH_4$ weight + $CO_2$ weight (4)

$$BMR = \frac{V_0 \times \left( \frac{16g}{1mole} \times \frac{CH_4}{100} + \frac{44g}{1mole} \times \frac{CO_2}{100} \right)}{\frac{22.413l}{mole}} \tag{5}$$

$$TVS_e = TVS_0 - BMR \tag{6}$$

## Kinetic modeling

The daily methane yield rate from BDCM during SSAD was simulated using the modified Gaussian and Gomperz equations (*Kim et al., 2018*). This model describes the daily methane yield in batch AD assuming that the kinetic growth of microbes including methanogens follows a normal distribution during the process. The Gaussian equation is presented as Eq. (7).

$$y = a \times \exp\left[ -0.5 \left( \frac{t - t_0}{b} \right)^2 \right] \tag{7}$$

y = Methane production rate (N mL/g VS) at any time t
t = digestion time (Day)
a = Constant (N mL/g VS/day)
b = Constant (Day)
$t_0$ = Maximal methane production rate day (Day)

The methane yield potential (P), maximum methane production (Rm; N mL/g VS), and lag phase ($\lambda$) were based on the volume and concentration of the biogas measured under the conditions of this experiment using the modified Gompertz equation (*Kafle & Chen, 2016*).

$$M = P \times \exp\left\{ -\exp\left[ \frac{R_m \times e}{P} (\lambda - t) + 1 \right] \right\} \tag{8}$$

M: Cumulative methane yield (N mL/g VS)
P: Methane yield potential (N mL/g VS)
$R_m$: Maximum methane production (N mL/g VS day)
$\lambda$: Lag phase
t: Day

## Statistical analysis

Data were analyzed using PROC MIXED of the SAS package program (SAS Inst. Inc., Cary, NC, USA) using retention time as a fixed model. Orthogonal polynomial contrasts were performed using the CONTRAST option to verify significance and polynomials were determined using SAS PROC IML. Statistical differences were considered significant at $p < 0.05$, and all averages were presented as least square means. Figures were created using GraphPad Prism 7.0 (GraphPad Software, San Diego, CA, USA).

## RESULTS

### Methane production, kinetic modeling, and methane content

The methane production on daily and cumulative basis during batch SSAD, and from the kinetic model using a modified Gompertz curve are shown in Fig. 1 and Table 3, respectively. The methane production from BDCM during batch SSAD increased rapidly from day 12 onwards with maximum yield of $19.6 \pm 0.8$ N mL/g VS/day, whereas maximum methane production was observed at $24.0 \pm 1.6$ days. The $R^2$ value of the Gaussian parameter using the modified Gompertz equation was 0.5992. Cumulative methane of BDCM was $142.5 \pm 6.0$ N mL/g VS at the end of SSAD. The maximum methane potential of BDCM was recorded as $134.7 \pm 9.9$ N mL/g VS according to the Gomperz curve and maximum methane production rate was $6.1 \pm 0.4$ N mL/g VS/day. The lag phase was $12.9 \pm 1.1$ days. The $R^2$ value of the Gompertz parameter using the modified Gompertz equation was 0.9373. The methane content of BDCM in the batch SSAD is shown in Fig. 2 with a maximum value of $76.1 \pm 7.3\%$. Results demonstrated that methane content increased rapidly until the 18th day, and it was maintained at $62.4 \pm 4.8\%$ until the end of the experiment. The $R^2$ value of the trend line obtained using the sigmoidal curve fitting was 0.9054.

### Ultimate biodegradability (UB) and volatile solid removal rate

The UB and VS removal rates of BDCM in the batch SSAD were found to be $23.1 \pm 3.4\%$ and $19.9 \pm 2.3$, respectively (Table 4). Moreover, the BVS removal rate was $86.4 \pm 1.4\%$.

### Fiber content removal

The results of cellulose, hemicellulose, and lignin contents of BDCM in SSAD are shown in Fig. 3. The cellulose and hemicellulose contents decreased linearly during SSAD ($p = 0.001$ and $p < 0.001$, respectively). In contrast, the lignin content increased ($p = 0.001$). The linear regression equations for cellulose, hemicellulose, and lignin content were as follows: cellulose (%, v. b.) $= -0.282\times$ fermentation time $+ 17.135$, hemicellulose (%, v. b.) $= -0.629\times$ fermentation time $+ 16.126$, and lignin (%, v. b.) $= 0.485\times$ fermentation time $+ 14.733$.

### Carbon and nitrogen removal

The carbon content of BDCM in the batch SSAD decreased quadratically during SSAD ($p < 0.001$; Fig. 4). The regression equations for non-fibrous carbon fraction (NFCF), hemicellulose carbon fraction (HCF), and acid detergent insoluble carbon (ADIC) were as follows: NFCF (%, v. b.) $= 0.001\times$ Fermentation time$^2$ - $0.1066\times$ fermentation time $+ 15.268$, HCF (%, v. b.) $= y = 0.0019\times$ Fermentation time$^2$ - $0.2172\times$ fermentation time $+ 16.125$, and ADIC (%, v. b.) $= y = 0.0001\times$ fermentation time$^2$- $0.0417\times$ fermentation time $+ 10.76$.

The nitrogen content of BDCM in the batch SSAD is shown in Fig. 5. Results revealed that the non-fibrous nitrogen fraction (NFNF) and hemicellulose nitrogen fraction (HNF) contents decreased quadratically ($p = 0.002$ and $p = 0.038$, respectively). In contrast, no significant decrease in acid detergent insoluble nitrogen (ADIN) content was observed

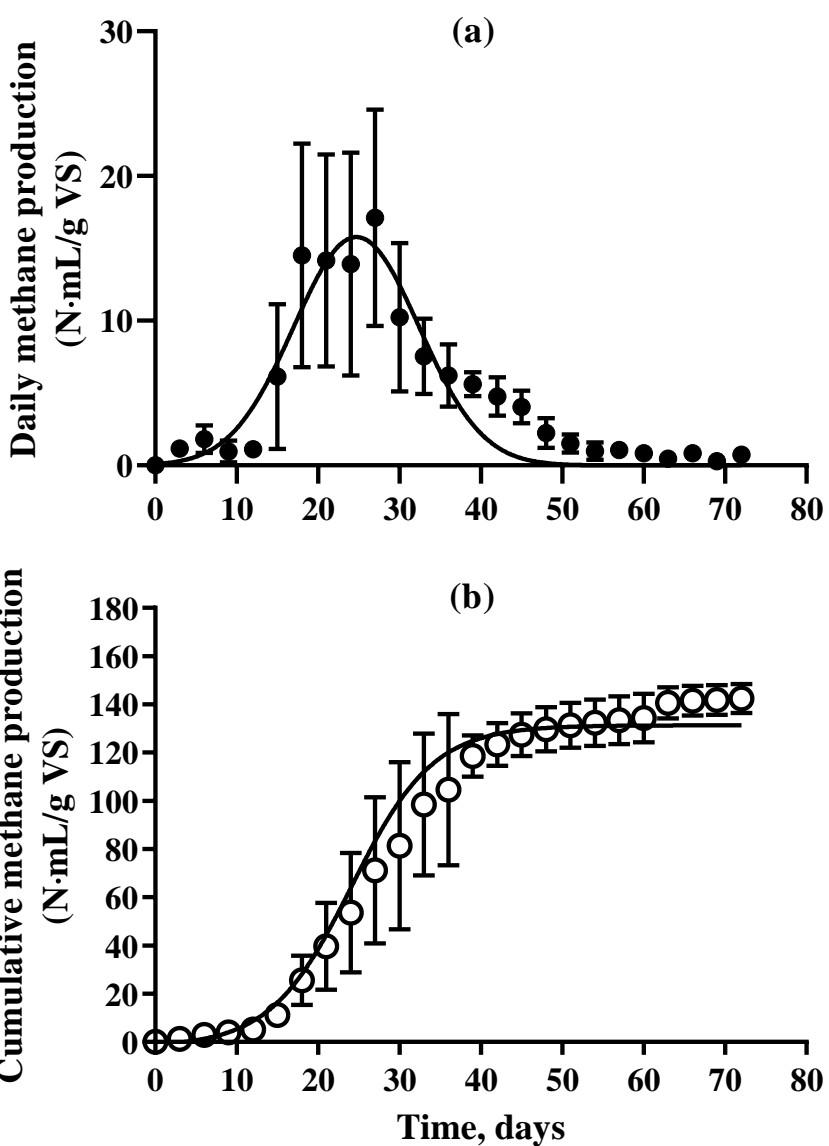

**Figure 1 Changes in the daily and cumulative methane production of bedded pack barn dairy cattle manure based on digestion time in the batch solid-state anaerobic digestion.** Changes in the daily and cumulative methane production of bedded pack barn dairy cattle manure based on digestion time in the batch solid-state anaerobic digestion ($n = 3$) (Table S1A). Daily methane production (A, ○) and cumulative methane production (B, *). Error bars indicate standard deviation.

($p = 0.836$). NFNF and HNF rapidly decreased until the day 12 and 6 of digestion, respectively ($p < 0.001$ and $p = 0.008$) but did not significantly decrease after day 15 until the end of the experiment. The regression equations of the NFNF, HNF, and ADIN were as follows: NFNF (%, v. b.) = $1.119 + (0.378/(1 + (\text{Fermentation time}/4.747)^{2.686})$), HNF (%, v. b.) = $0.268 + (0.431/(1 + (\text{Fermentation time}/2.737)^{19.421})$), and ADIN (%, v. b.) = $0.319 + (-0.046 / (1 + (\text{Fermentation time}/20.489)^{122.57}))$.

**Table 3** Gaussian and Gompertz curve of methane production in bedded pack barn dairy cattle manure in the batch solid-state anaerobic digestion ($n = 3$).

| Parameter[a] | BDCM (Mean ± SD) |
| --- | --- |
| Gaussian curve | |
| Maximum specific methane yield (N mL/ g VS / day) | 19.5 ± 0.8 |
| $t_0$ | 24.0 ± 1.6 |
| a (N mL/ g VS /day) | 19.6 ± 0.8 |
| b (days) | 7.5 ± 0.7 |
| R square | 0.5992 |
| Gomperz curve | |
| Cumulative methane yield (N mL/ g VS) | 142.5 ± 6.0 |
| $P^1$ (N mL/ g VS) | 134.7 ± 9.9 |
| $R_m^2$ (N mL/ g VS/day) | 6.1 ± 0.4 |
| $\lambda^3$ (days) | 12.9 ± 1.1 |
| R square | 0.9263 |

**Notes.**

BDCM, Bedded pack barn dairy cattle manure; SD, standard deviation.

[a]$t_0$, the days of maximum methane yield; a, daily maximum methane yield (constant); b, days (constant); P, cumulative methane potential; $R_m$, maximum daily methane production rate; $\lambda$, lag phase.

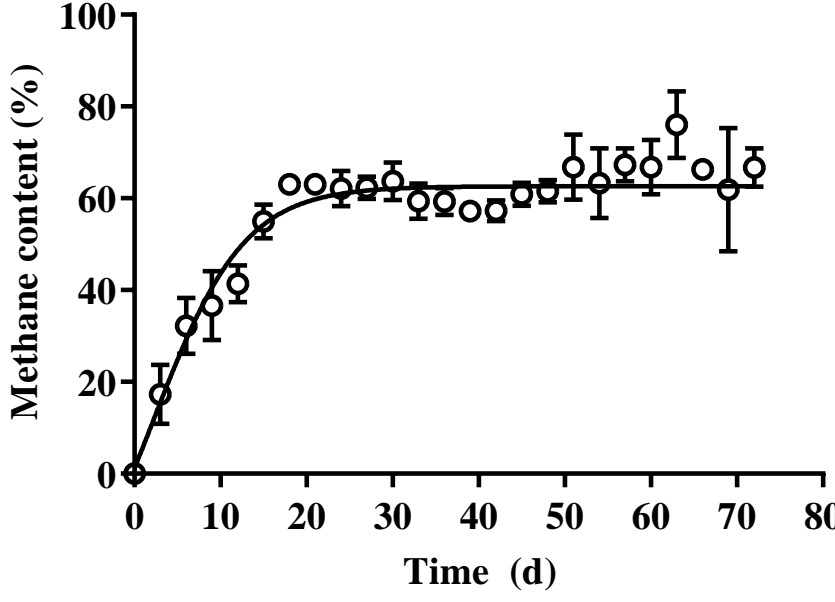

**Figure 2** Changes in the methane content of bedded pack barn dairy cattle manure based on digestion time in the batch solid-state anaerobic digestion. Changes in the methane content of bedded pack barn dairy cattle manure based on digestion time in the batch solid-state anaerobic digestion ($n = 3$) (Table S2A). Error bars indicate standard deviation.

## pH, NH₃, NH₄-N, and volatile fatty acids

Results demonstrated a change in the pH of BDCM ranging from 7.5–8.0 in batch SSAD until day 12 of fermentation. However, it increased to 8.5 and was maintained at 8.0 thereafter (Fig. 6).

**Table 4** Ultimate biodegradability and volatile solid removal on bedded pack barn dairy cattle manure in the batch solid-state anaerobic digestion ($n = 3$).

| Item | Initial VS (g) | BMR (g) | UB (%) | TVS removal (%) | BVS removal[1] (%) |
|------|----------------|---------|--------|-----------------|--------------------|
| DCM | $38.6 \pm 0.7$ | $7.7 \pm 0.9$ | $23.1 \pm 3.4$ | $19.9 \pm 2.3$ | $86.4 \pm 1.4$ |

**Notes.**

BMR, biomass removal; UB, ultimate biodegradability; TVS, total volatile solids; BVS, biodegradable volatile solids; DCM, the bedded pack barn dairy cattle manure.

[a]Based on UB of each fraction.

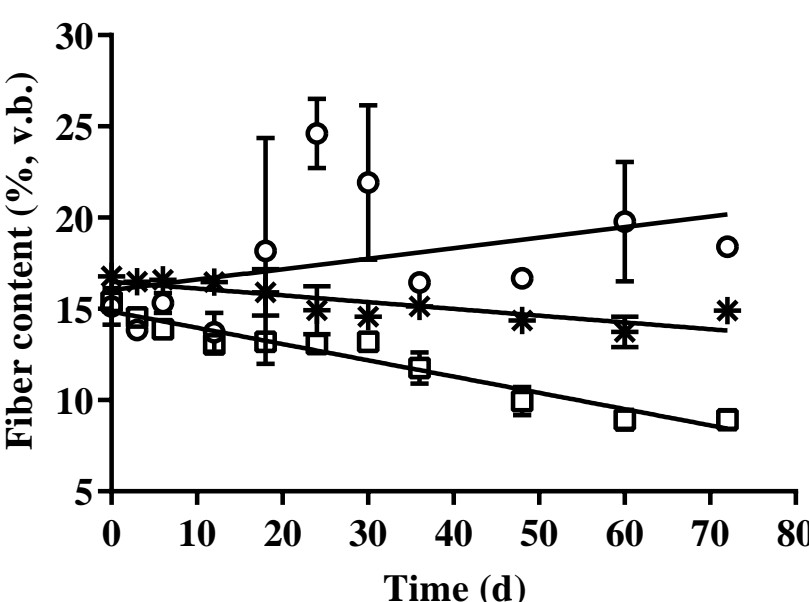

**Figure 3** **Changes in the cellulose, hemicellulose, and lignin content of bedded pack barn dairy cattle manure according to digestion time in the batch solid-state anaerobic digestion.** Changes in the cellulose, hemicellulose, and lignin content of bedded pack barn dairy cattle manure according to digestion time in the batch solid-state anaerobic digestion ($n = 3$) (Table S3A). Cellulose (*), hemicellulose (□), and lignin (○). v.b., Volatile solid basis. Error bars indicate standard deviation.

The ammonia nitrogen ($NH_4$-N) content and maximum free ammonia ($NH_3$) concentration of BDCM was maintained at 1,200 mg/L and $147.7 \pm 13.8$ mg/L during the entire batch SSAD (Fig. 7). However, the $NH_3$ concentration decreased to <100 mg/L after 18th day.

Results of VFA concentration of BDCM in the SSAD batch are shown in Fig. 8, and revealed that the maximum acetic, propionic, and VFA concentrations were $223.0 \pm 10.8$, $110.5 \pm 11.9$, and $357.3 \pm 21.5$ mg/L, respectively. Moreover, VFA concentrations were mostly maintained after 36 days of batch SSAD.

## DISCUSSION

The cumulative methane production of BDCM in the batch SSAD was 127.7 N mL/g VS after 45 days of fermentation, with a methane potential of 95.0% according to the

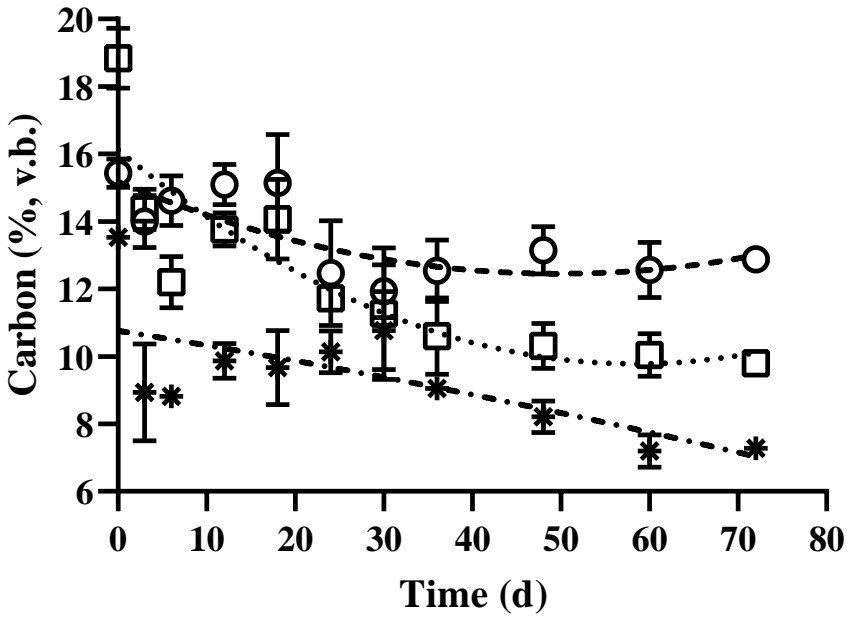

**Figure 4  Changes in the carbon content of bedded pack barn dairy cattle manure based on digestion time in the batch solid-state anaerobic digestion.** Changes in the carbon content of bedded pack barn dairy cattle manure based on digestion time in the batch solid-state anaerobic digestion ($n = 3$) (Table S4A). ○, Non-fibrous carbon fraction (quadratic equation, $y = 0.001 \times$ fermentation time$^2$ - 0.1066 × fermentation time + 15.268). □, Hemicellulose carbon fraction (quadratic equation, $y = 0.0019 \times$ fermentation time$^2$ - 0.2172 × fermentation time + 16.125); *, acid detergent insoluble carbon (cellulose + lignin; quadratic equation, $y = 0.0001 \times$ fermentation time$^2$ - 0.0417 × fermentation time + 10.76); v.b., volatile solid basis. Error bars indicate standard deviation.

kinetic model. This observation differed from previous reports (*Triolo et al., 2011*; *Kim et al., 2018*). Additionally, the average methane content was found to be 62.4% from day 18 to 72. However, according to previous studies, it ranges from 65 to 80% (*Rosas-Mendoza et al., 2021*, p.), or 40 to 55% during AD process (*Wang et al., 2012*; *Goux et al., 2016*). Further, our results demonstrated no change in the methane content of BDCM in the batch SSAD after a digestion time of 18–72 days. Moreover, no reduction in methane content also suggested that methane production was not inhibited during the AD process. According to a previous study (*Triolo et al., 2011*), cattle manure with similar contents of fibrous materials as used in our study, was able to produce 197–223.6 N mL/g VS of methane. Although the present study used similar substrates during SSAD, the moisture conditions, lignin content of feeding substrates, and substrate to inoculum ratio were different. The moisture conditions in the previous study were 89.6 and 92.5%, whereas our study reported it as 82%. Water activity is known to affect the microbial growth during the digestion process, and the optimum water activities may vary among microbial species (*Gervais, Marechal & Molin, 1996*). Methane production was the highest at 85% moisture in our study, and higher production occurred at 82% moisture than at 89.6 and 92.5% (*Veluchamy & Kalamdhad, 2017*). These findings suggest that moisture condition was not a factor associated with decreased methane production in the present study. Furthermore,

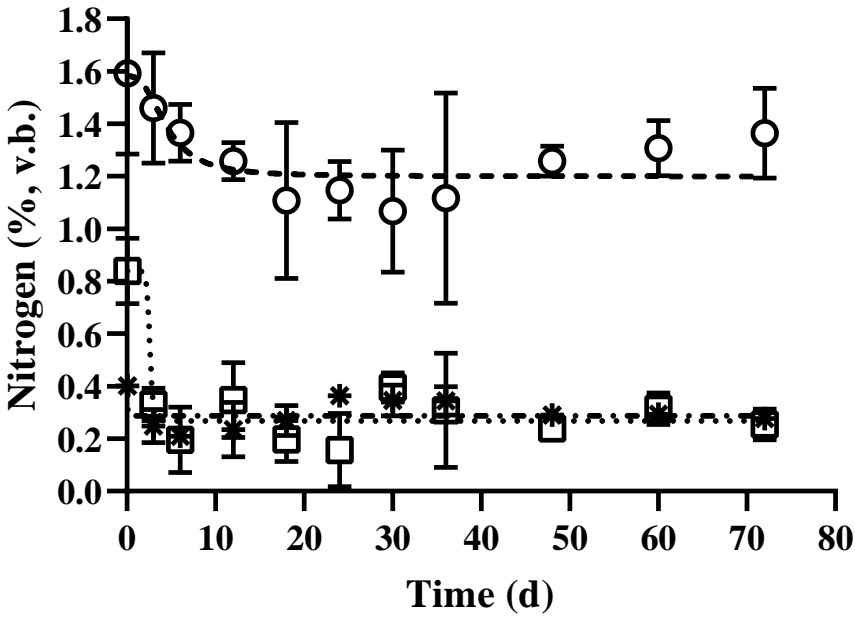

**Figure 5 Changes in the nitrogen content of bedded pack barn dairy cattle manure based on digestion time in the batch solid-state anaerobic digestion.** Changes in the nitrogen content of bedded pack barn dairy cattle manure based on digestion time in the batch solid-state anaerobic digestion ($n = 3$) (Table S5A). ◯, Non-fibrous nitrogen fraction (Sysmetrical sigmoidal equation, $y = 1.119 + (0.378/(1 + (\text{Fermentation time}/4.747)^{2.686})$; ☐, hemicellulose nitrogen fraction (Sysmetrical sigmoidal equation, $y = 0.268 + (0.431/(1 + (\text{Fermentation time}/ 2.737)^{19.421})$; *, acid detergent insoluble nitrogen (cellulose + lignin; [Sysmetrical sigmoidal equation, $y = 0.319 + (-0.046/(1 + (\text{Fermentation time}/20.489)^{122.57}))])$; %, volatile solid basis. Error bars indicate standard deviation.

our results reported lignin content of feeding substrate as 17.5% of dry matter (DM); however, previous studies have reported it as 29.1% (*Kim et al., 2018*) and 6.49–11.91% of DM (*Triolo et al., 2011*). Correspondingly, the methane production was 142.5 (our study), 90.3 (*Kim et al., 2018*), and 197.0–223.6 N mL g/g VS (*Triolo et al., 2011*). Increasing lignin content in feeding substrates causes a linear decrease in methane production and biochemical methane potential (BMP) (*Triolo et al., 2011*; *Choi et al., 2020*). Lignin is the primary component of recalcitrant fraction of lignocellulose (*Khan & Ahring, 2019*), and its biological conversion is limited in the absence of oxygen; however, anaerobic conditions are necessary for methane production by biological digestion (*Harun et al., 2010*). Additionally, sawdust is a by-product of woodworking such as sawing, sanding, routing, *etc.*, and lignin constitutes 81.59% of VS of the pinewood sawdust (*Muley et al., 2016*). When the VS content was calculated excluding lignin content of substrates, it was found to be 85.4–88.4 (*Triolo et al., 2011*), 70.9 (*Kim et al., 2018*), and 85.5% of VS (our study). Therefore, the lignin content of substrates was a significant factor affecting methane production. Nevertheless, a huge difference in methane production was reported between the present study (142.5 N mL g/g VS) and that of *Triolo et al. (2011)* (197.0–223.6 N mL g/g VS). The substrate to inoculum (S/I) ratio has also been reported to affect the cumulative methane production which was the highest at S/I ratio 1.0; however, it was found to be
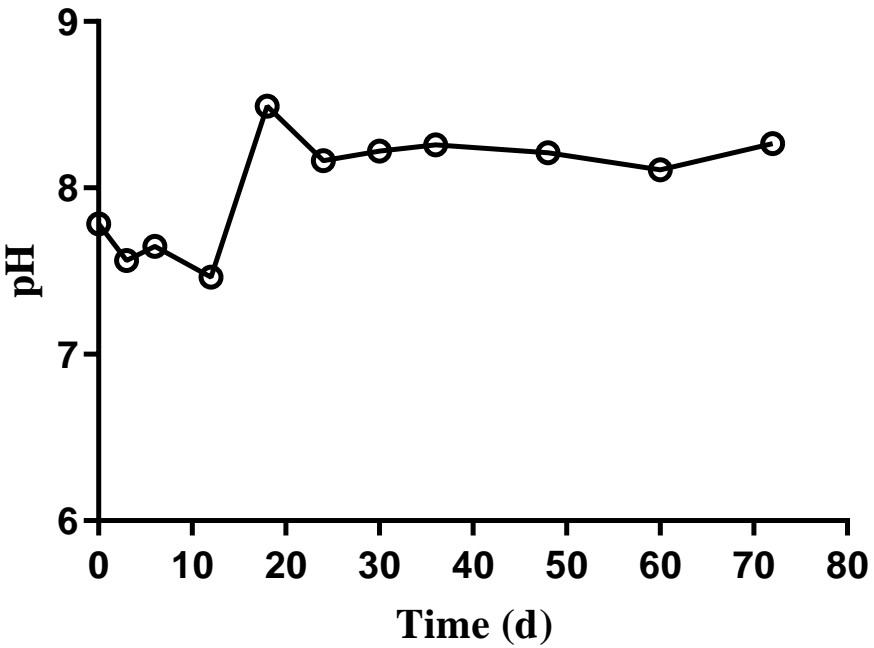

**Figure 6** **Changes in the pH of bedded pack barn dairy cattle manure based on digestion time in the batch solid-state anaerobic digestion.** Changes in the pH of bedded pack barn dairy cattle manure based on digestion time in the batch solid-state anaerobic digestion ($n = 3$). Error bars indicate standard deviation.

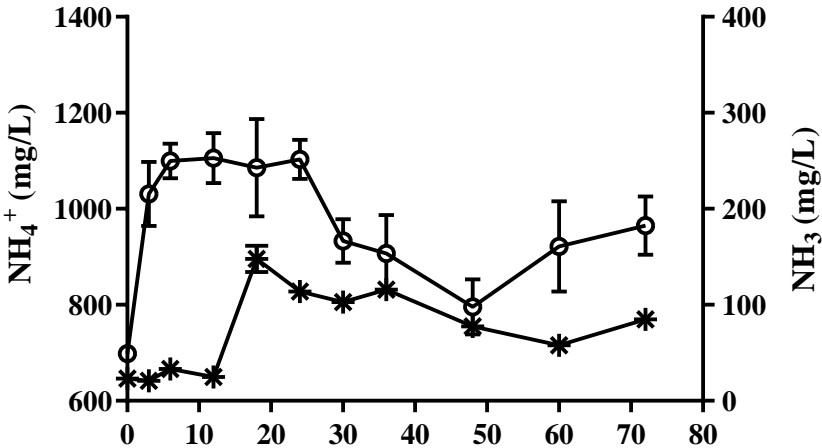

**Figure 7** **Change in ammonia nitrogen and free ammonia content of bedded pack barn dairy cattle manure according to digestion time in batch solid-state anaerobic digestion.** Change in ammonia nitrogen and free ammonia content of bedded pack barn dairy cattle manure according to digestion time in batch solid-state anaerobic digestion ($n = 3$). Ammonia nitrogen ($\bigcirc$, $NH_4$-N), free ammonia ($*$, $NH_3$). Error bars indicate standard deviation.

Choi et al. (2022), *PeerJ*, DOI 10.7717/peerj.14134 12/19

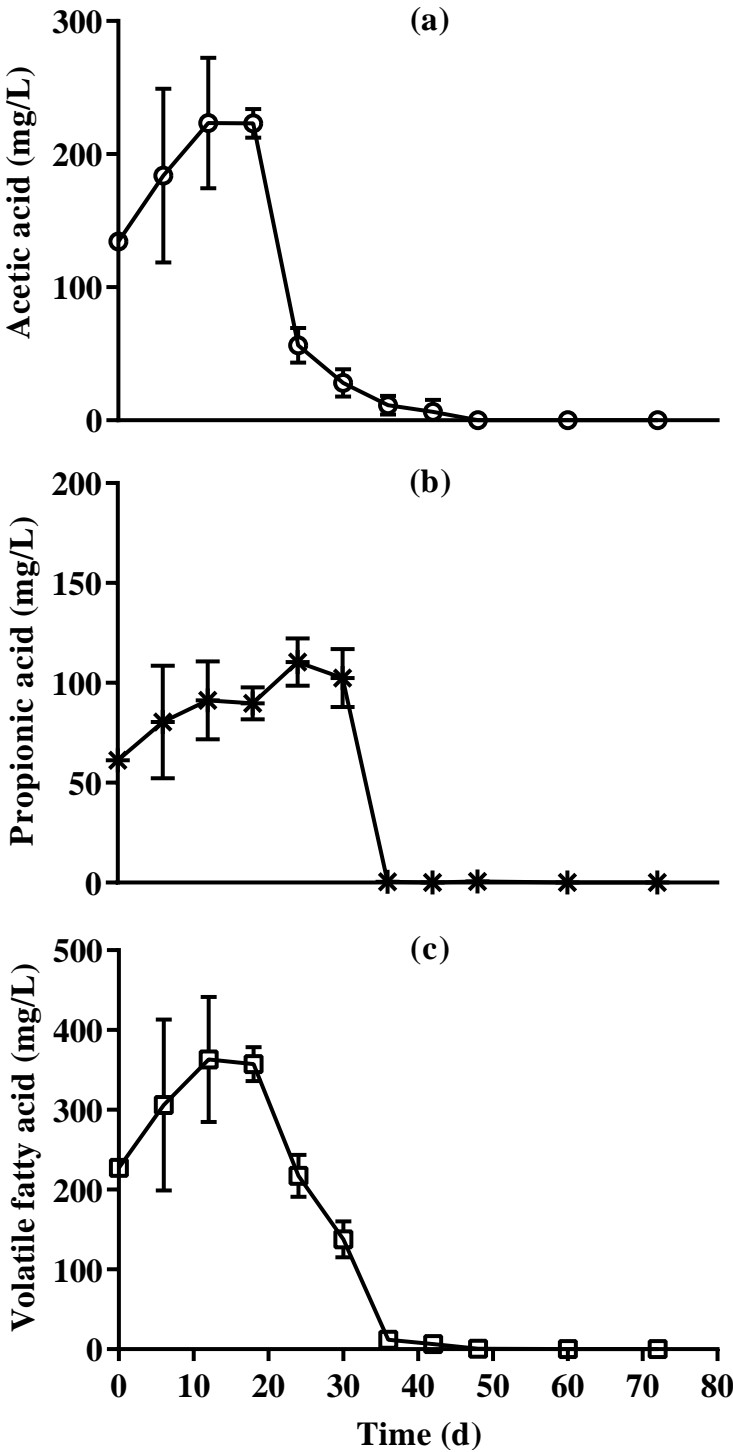

**Figure 8** **Changes in acetic acid, propionic acid , and total volatile fatty acid content reduction of bedded pack barn dairy cattle manure based on the batch solid-state anaerobic digestion.** Changes in acetic acid, propionic acid , and total volatile fatty acid content reduction of bedded pack barn dairy cattle manure based on the batch solid-state anaerobic digestion ($n = 3$). Acetic acid (A, ○), propionic acid (B, *), and total volatile fatty acid (C, □). Error bars indicate standard deviation.

lower (approximately half) at other S/I ratios (0.25, 1.5, and 2.0) (*Haider et al., 2015*). The study by *Triolo et al. (2011)* was conducted at S/I ratio of 1.0 while we performed it with 0.05. Therefore, the difference observed in cumulative methane production may have resulted from varying S/I ratios. The UB and VS removal rates in our study were found to be lower than those reported in previous studies (*Khalid et al., 2011*). Furthermore, as UB and VS removal rates are calculated using methane and carbon dioxide yields, the lower values may be a result of low methane yield.

The equation of fibrous contents is presented to evaluate the degradation of each fibrous fraction. Results revealed that the hemicellulose and cellulose contents of BDCM decreased during SSAD, while the lignin content increased. Unmodified lignin is an undegradable component of the AD process (*Khan & Ahring, 2019*). In this study, the cellulose content was calculated by subtracting the lignin from acid detergent fiber content, and the hemicellulose content was calculated by subtracting the acid detergent fiber from the neutral detergent fiber content. Therefore, increase in lignin content with fermentation progress may have resulted from the decomposition of cellulose and hemicellulose. In addition, the fibrous content showed that cellulose degraded more slowly than hemicellulose (Fig. 3). Previous reports showed that xylan, which is hemicellulose, degrades faster than starch and cellulose during the BMP test (*Choi, Ryu & Lee, 2020*). Our study also demonstrated the same results. Therefore, hemicellulose is considered to be the most rapid degradable fraction of fibrous materials during SSAD.

In this study, carbon and nitrogen contents were classified under non-fibrous, hemicellulose, and acid detergent insoluble fractions including cellulose and lignin fractions according to AOAC method (*AOAC, 2005*). In general, this classification is widely used in ruminant feed studies; however, it exhibited different utilization patterns than the ruminant classifications similar to AD (*NRC, 2001*). According to a previous study, carbon sources are classified by carbon bonds, which affect the AD patterns of the feeding materials (*Choi, Ryu & Lee, 2020*). The regression equations were used to determine the patterns of decarbonization and denitrification during SSAD, which correspond to the degradation characteristics of carbon and nitrogen of feeding substrates. All the equations were selected after evaluation of data with linear and quadratic effects before equation fitting. In contrast, a sigmoidal equation (NFCF and HFCF) or constant pattern (ADIN) during SSAD was followed by nitrogen fractions. In this study, a quadratic pattern or decreasing degradation of carbon and nitrogen was observed with AD progression. The NFCF and HFCF showed a decrease in degradability from days 40 to 50 of the AD process, and the ADIC group showed a slower decrease in overall digestion. However, NFNF and HFNF indicated that digestibility of nitrogen almost stopped during the early AD process in about 6–12 days, and ADIN showed that it was hardly utilized during the entire AD process. Normally, in AD studies, nitrogen is analyzed in the whole feedstock for the calculation of the C/N ratio in the feeding material. According to previous reports, methane was maximally produced in the C/N ratio of 20–35 (*Puñal et al., 2000*; *Yan et al., 2015*). Moreover, ADIN is rarely degraded by microbes in the rumen (*Machacek & Kononoff, 2009*); therefore, it is difficult to decompose during AD process which is similar to rumen. However, in batch AD, nitrogen undergoes decomposition at an early stage to support microbial growth. Nitrogen

removal is reported to be efficient under aerobic conditions, and the total Kjeldahl nitrogen is constant in continuous AD (*Novak, Banjade & Murthy, 2011*). In this study, since ADIN rarely degraded during the entire AD process for 72 days, nitrogen availability can be precisely calculated by removing undegradable nitrogen from the ADIN.

Although a pH of over 8.0 was reported that inhibited the AD (*Cioabla et al., 2012*), no inhibition due to pH change was observed during the entire batch SSAD process. In AD process, the inhibition of ammonia nitrogen ($NH_4$-N) and free ammonia ($NH_3$) was known to occur at over 3,000 and 80 mg/L, respectively (*Yenigün & Demirel, 2013*). In our study, $NH_4$-N content was under 1,200 mg/L. Serious inhibition has been reported at a pH of 8.0 and at over 100 mg/L of $NH_3$ in the AD process; however, no inhibition possibly occurred in this study as the $NH_3$ concentration was under 100 mg/L after 18 days. The inhibition by VFA accumulation has been reported to occur at over 2,400 (*Wang et al., 2009*), 2,750, (*Drosg, 2013*), and 4,000 mg/L (*Lee et al., 2015*). The results of this study suggest that the VFA content does not inhibit the entire SSAD process.

Overall, the BDCM used in this study was found to have low biodegradability, owing to the incorporation of sawdust. In addition, digestion proceeded normally without any inhibitory action during the entire SSAD process. Further, differences in carbohydrate structure of the digested material in the batch SSAD and ADIN were confirmed to make degradation difficult during the AD process. Therefore, this study provides information on BDCM digestion characteristics in the SSAD which can be of significant help in improving the SSAD process.

## CONCLUSIONS

This study evaluated the digestion characteristics of BDCM in batch SSAD. The cumulative methane production at the end of SSAD was $142.5 \pm 6.0$ N mL/g VS. The average methane content of BDCM during batch SSAD was 62.4% from days 18 to 72. The TVS and BVS removal rates during the batch SSAD were $19.9 \pm 2.3\%$ and $86.4 \pm 1.4\%$, respectively. The carbon fraction of NFCF, HCF, and ADIC is continuously degraded during the entire anaerobic digestion unlike the nitrogen fraction of NFNF and HNF, which is only degraded in early AD, and that of ADIN is rarely degraded. Consequently, this study provides information on BDCM digestion characteristics in SSAD, which will help in improving the SSAD process as basic data.

### Funding

This research was supported by the Basic Science Research Program through the National Research Foundation of Korea (NRF) funded by the Ministry of Education (2021R1I1A1A01058751) and the 2018 Post-Doc Department of Animal Resources Technology, Gyeongnam National University of Science and Technology, and Agri-Food Bio Convergence Institute. The funders had no role in study design, data collection and analysis, decision to publish, or preparation of the manuscript.

## Grant Disclosures

The following grant information was disclosed by the authors:
Ministry of Education: 2021R1I1A1A01058751.
2018 Post-Doc Department of Animal Resources Technology.
Gyeongnam National University of Science and Technology.
Agri-Food Bio Convergence Institute.

## Competing Interests

The authors declare there are no competing interests. Youngjun Na is an employer (chief executive officer) of Animal Data Lab., Antller Inc.

## Author Contributions

- Yongjun Choi conceived and designed the experiments, performed the experiments, analyzed the data, prepared figures and/or tables, authored or reviewed drafts of the article, and approved the final draft.
- Sangrak Lee conceived and designed the experiments, prepared figures and/or tables, authored or reviewed drafts of the article, and approved the final draft.
- Duck-Min Ha performed the experiments, prepared figures and/or tables, and approved the final draft.
- Youngjun Na analyzed the data, authored or reviewed drafts of the article, and approved the final draft.
- Doo-Hwan Kim conceived and designed the experiments, authored or reviewed drafts of the article, and approved the final draft.

## Data Availability

All raw data are available in the Supplementary Files.

## Supplemental Information

Supplemental information for this article can be found online at http://dx.doi.org/10.7717/peerj.14134#supplemental-information.

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
