# Peer review of "Fermentation characteristics of bedded pack barn dairy cattle manure on methane yield, carbon, and nitrogen content in solid-state anaerobic digestion"

_PeerJ, doi:10.7717/peerj.14134_

## Round 0.1 · original submission · Major Revisions

Your manuscript 'Fermentation characteristics of bedded pack barn dairy cattle manure on methane yield, carbon, and nitrogen content in solid-state anaerobic digestion ' has been assessed by our reviewers. They have raised a number of points which we believe would improve the manuscript and may allow a revised version to be considered for publication in PeerJ. If you are able to fully address these points, we would encourage you to submit a revised manuscript.

Reviewer 1 ·

Basic reporting

However, the overall research content was relatively shallow, and there was no discussion on some reasons. It was only a statement of the results. Although a significant analysis of mathematical correlation was carried out, the overall research depth was not enough.
It is unclear why sawdust in BDCM reduces methane production.

Experimental design

no comment

Validity of the findings

1. Discussion part analysis disorder, no reason, not concise ;
2. The conclusion is too simple, not clear, the suggestion should be more specific and clear ;

Additional comments

1. Decarbonization and denitrification only carried out the calculation of the regression equation, and did not conduct more in-depth analysis, did not clearly explain the exact significance of the regression equation ;
2. General representation of graphs
3. The format needs to be slightly improved

Annotated reviews are not available for download in order to protect the identity of reviewers who chose to remain anonymous.

Reviewer 2 ·

Basic reporting

This reviewer commends the authors’ effort in the presentation of the study. The language used was professional throughout the manuscript, and the structure conformed to PeerJ standards. The figures and tables are also relevant to the study. However, this reviewer found a few sentences slightly vague and unclear. Furthermore, this reviewer opines that the introduction did not adequately capture the novelty of the study or show an in-depth review of the literature. The cited literature was well-referenced; however, there were several archaic references—especially in the methods. Please refer to the “general comments’ below for additional information.

Experimental design

This reviewer finds the study interesting and highly relevant. However, the author’s primary justification for the study was that the study “provides information…which could help improve the SSAD process.” This reviewer believes the authors can improve the manuscript by clarifying and emphasizing the impact and novelty of the study. This reviewer also believes that all underlying data have been provided and are statistically sound. The conclusions are also well stated. However, the inclusion of equations in the conclusion may not be necessary.

Validity of the findings

This reviewer determines that the original primary research is within the scope of the journal, and the authors performed the investigation to a high technical & ethical standard. However, the research question was not well-defined, and the authors did not adequately present how the research fills an identified knowledge gap. Although the methods are described with sufficient detail and information to replicate, this reviewer believes the manuscript can be significantly improved by citing more recent studies that used similar methods, as opposed to the studies conducted in 1962, 1979, 1991, etc.

Additional comments

Please consider the following general comments:

1. Line 37: The sentence is a very strong opening statement showing the superiority of one method over another. Should this be supported by references?
2. Lines 37–39: What number is not large, and what is the connection between conducting AD in wet state and the number?
3. Line 39–40: The previous sentence stated “In South Korea…;” this sentence stated “…in European countries…;” and the next statement stated “in South Korea…” Should these sentences be revised to either focus on the main area of the study, or provide a broader, global perspective?
4. Line 43: Please delete one “the.”
5. Line 44: “…more disposed of…” is slightly vague. Please consider clarifying this phrase.
6. Lines 49¬–51: Previous studies such as Kim et al., 2018 (https://www.mdpi.com/266440) have conducted some relevant studies using similar materials. Please add more information on previous studies on SSAD of manure and sawdust before drawing the conclusion that studies are rare.
7. Lines 52–56: Please revise the first sentence for grammatical errors. Also, the authors stated two types of AD but only presented the information on the batch process. Should the continuous process be presented to provide a balanced perspective?
8. Lines 57–58: Please add why the study was necessary and what is novel about the study. What knowledge gap is the study filling, other than providing some reference information?
9. Lines 93, 97, 104, 125, and 132: Using only old references suggests a lack of in-depth literature review. Are there any revisions of the methods, or other studies that have adopted the methods over the years/decades since the cited works were published? Please consider adding more recent references or replacing the old references with newer, more recent citations.
10. Figure 1a: Please consider adding a label and scale to the x-axis, for clarity. Also, please consider adding an explanation for the wide variability in the data after day 6 (or the sixth data point on the chart).
11. Lines 178–179: Please check/verify the sentence on the non-linear decrease of the cellulose content of BDCM. The chart/trendline seems to suggest otherwise. If the sentence is true, then some clarifying statement may be necessary.
12. Line 223: “it was reported that animal manure…” Please specify which animal.
13. Line 225: Should the sentence be: “A previous study showed that…”?
14. Lines 227–228: the sentence “…which suggests that the difference in methane production between the two studies was affected.” Is unclear. Please clarify.
15. Line 229: Please clarify which previous study. Also, the authors compared results with only one previous study (Triolo et al., 2011). Are there other previous/relevant studies? Comparing the results with additional relevant studies will improve the quality of the manuscript and ensure a more in-depth knowledge of previously completed studies on the subject matter.
16. Lines 305–308 and 310–313: Is it necessary to include the regression equations in the conclusion? Please consider removing the equations and, rather, summarize the implication of the equations.

---

## Round 0.2 · Minor Revisions

Your manuscript requires a number of Minor Revisions.

Reviewer 1 ·

Basic reporting

no comment

Experimental design

no comment

Validity of the findings

no comment

Additional comments

no comment

Reviewer 2 ·

Basic reporting

This reviewer commends the authors’ effort in addressing the comments previously raised by the reviewers. In addition, the authors replaced the older references/citations with more recent ones, provided better background/context than in the initial submission, and updated the equations and discussions. Therefore, this reviewer believes that the revised manuscript is an improved version. However, the quality of the current version of the manuscript is significantly weakened by the number of grammatical errors. Therefore, a review of the manuscript for grammar is strongly recommended. If possible, the authors should recruit a fluent English speaker to help.

Experimental design

The experimental design appears satisfactory.

Validity of the findings

This reviewer commends the edits and updates made by the authors. However, given the significance of the mistakes discovered during the review process, another review of the data analysis/interpretation is highly recommended.

---

## Round 0.3 · accepted · Accept

Dear Dr. Choi,

Thank you for your submission to PeerJ.

I am writing to inform you that your manuscript - Fermentation characteristics of bedded pack barn dairy cattle manure on methane yield, carbon, and nitrogen content in solid-state anaerobic digestion - has been Accepted for publication. Congratulations!